# Analysis on Water Inrush Process of Tunnel with Large Buried Depth and High Water Pressure

**Weimin Yang [1,2], Zhongdong Fang [1,2], Hao Wang [1,3], Liping Li [1,2,*], Shaoshuai Shi [1,2,*]**, **Ruosong Ding [1,2], Lin Bu [1,3] and Meixia Wang [1,3]**

1   Geotechnical and Structural Engineering Research Center, Shandong University, Jinan 250061, China; weimin.yang@sdu.edu.cn (W.Y.); fzd0208@mail.sdu.edu.cn (Z.F.); wanghao15@mail.sdu.edu.cn (H.W.); 201734617@mail.sdu.edu.cn (R.D.); 201520363@mail.sdu.edu.cn (L.B.); 201613310@mail.sdu.edu.cn (M.W.)
2   School of Qilu Transportation, Shandong University, Jinan 250061, China
3   School of Civil Engineering, Shandong University, Jinan 250061, China
*   Correspondence: shishaoshuai@sdu.edu.cn (S.S.); liliping@sdu.edu.cn (L.L.); Tel.: +86-531-88399080 (S.S. & L.L.)

**Abstract:** In order to explore the catastrophic evolution process for karst cave water inrush in large buried depth and high water pressure tunnels, a model test system was developed, and a similar fluid–solid coupled material was found. A model of the catastrophic evolution of water inrush was developed based on the Xiema Tunnel, and the experimental section was simulated using the finite element method. By analyzing the interaction between groundwater and the surrounding rocks during tunnel excavation, the law of occurrence of water inrush disaster was summarized. The water inrush process of a karst cave containing high-pressure water was divided into three stages: the production of a water flowing fracture, the expansion of the water flowing fracture, and the connection of the water flowing fracture. The main cause of water inrush in karst caves is the penetration and weakening of high-pressure water on the surrounding rock. This effect is becoming more and more obvious as tunnel excavation progresses. The numerical simulation results showed that the outburst prevention thickness of the surrounding rock is 4.5 m, and that of the model test result is 5 m. Thus, the results of the two methods are relatively close to each other. This work is important for studying the impact of groundwater on underground engineering, and it is of great significance to avoid water inrush in tunnels.

**Keywords:** large buried depth; high water pressure; karst cave water inrush; catastrophic evolution; model test

## 1. Introduction

Solution fissures develop in karst areas, and it is easy to form a water-filled karst cave in a soluble rock formation or an area with more runoff. An inrush of clay and water easily occurs in tunnels near a karst cave. As the integrity of the rock mass is poor, the strength of the rock mass is low, and therefore, it contains more water flowing fractures, endangering the safety of the construction workers, and causing project delays and economic losses [1–4]. As shown in Figure 1, water inrush occurs in a tunnel. The determination of the safe thickness between the tunnel and the surrounding caves, which mainly depends on the construction method and design technology, lacks a reference system theory or technology [5,6]. In addition, each type of hidden cave responds differently to instability and breakdown [7–9]. Therefore, there is important practical significance and engineering value to studying the bursting process and mechanism of a catastrophe evolution of a typical water inrush type in a karst cave [10–12].

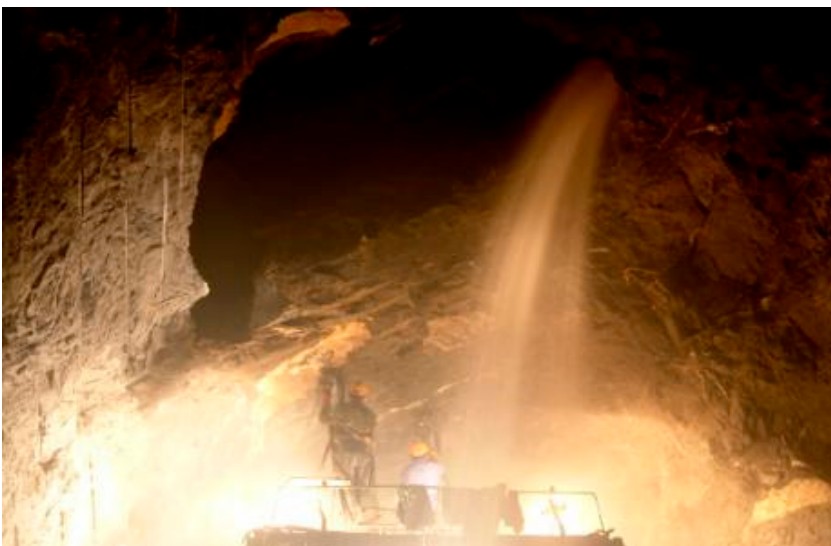

**Figure 1.** Water inrush in a tunnel.

At present, scholars are deeply studying the inrush of clay and water in karst caves through various methods in terms of the distribution of the karst structure and the properties of the filling materials. On the theoretical side, Zhu et al. considered the actual hydraulic behaviors to be a combination of the long-term trends and short-term effects, and proposed a prediction method called a 'time window' for the risk of a lagging water inrush [13]. Hu et al. established instability models for the filling of a karst conduit and obtained the corresponding instability criteria of these models by deriving their mechanics formulas. Based on karst water inrush in the Fenshuiao tunnel project, the mechanism and numerical simulation results of its water inrush process were analyzed [14]. Zhao et al. used a linkage analysis between the hydro-mechanical coupling and a strength reduction method to investigate the stability of water-resistant rock pillars [15]. On the numerical simulation side, based on the damage mechanism theory, Shan et al. used FLAC$^{3D}$ software to simulate and analyze the different locations of karst caves. The influencing factors, including the safety thickness and failure mechanism, of the surrounding rocks of tunnels and hidden karst caves in the Cheng-gui railway were analyzed [16]. Zhou et al. established a typical calculation model for an underground stope of a concealed fault by applying the finite element method and fracture mechanics method, and quantitatively evaluated the degree of stress concentration. The mining depth, fluid pressure, fault dip angle, and fault length were analyzed, and the working face was simulated to determine whether the underground mining will cause a fault activation [17]. On the model test side, Zhang et al. conducted a physical simulation of the fracture formation, propagation of buried faults, and evolution of the water inrush channel under high-pressure water in a coal mine floor on the basis of solid material research and fluid–solid coupling mechanics, and revealed the mechanism of water inrush from a coal seam floor in confined water under pressure and seepage fields [18]. Li et al. built a three-dimensional (3D) model test and conducted a numerical simulation for an undersea tunnel with a filling type fault to analyze the mechanism of water inrush, and studied the changes in the excavation disturbed zone during excavation in the undersea tunnel [19]. The displacement and hydraulic pressure in the tunnel construction process are often interactive as both cause and effect. The ongoing maturation of the geomechanical model test technique provides a new channel to research the catastrophic evolution mechanism and the precursor information changes the rules of water inrush during tunnel construction.

The above studies investigated the causes of disasters from the aspects of fracture expansion, pressure flow state conversion, and so on, or the catastrophic evolution based on the structure of the water and the properties of the fillings, which lack regularity and a systematic analysis regarding the change in the surrounding rock physical information of karst cave water inrush. In the process of

tunnel excavation, the existence status of the disaster source is not clear. The karst cave water inrush is a process of rapid pressure release after the pressurized water in the cavern penetrates the disturbed rock mass. Affected by the excavation disturbance, the cracks inside the aquifuge rock expand and the water inrush channel is formed, which eventually leads to disasters. At present, the mechanism of this type of water inrush is still unclear.

This work is based on a specific engineering example. By analyzing the interaction between groundwater and the surrounding rock during tunnel excavation, the law of occurrence of water inrush disaster is summarized. This work provides technical support for the reservation of the minimum safe thickness in tunnel construction. Model tests and numerical simulation studies at work are adopted. The mutual verification of the two results guarantees its reliability. This work is important for studying the impact of groundwater on underground engineering, and it is of great significance for avoiding water inrush in tunnels.

First, the engineering background is presented. Then, the process of the model test is introduced. The numerical simulation process is presented next. Finally, the results of the two research methods are comprehensively analyzed, and the corresponding conclusions are drawn.

## 2. Engineering Background

The Xiema Tunnel is a control project for the "one horizontal line" in the Chongqing Expressway system. The tunnel starts from Xiema Town in Beibei District, passing through the Zhongliang Mountain Range, and east to Caijiagang Town in Beibei District. The road tunnel is designed as a split twin-tube tunnel. There are three lanes in one direction. The construction method is the benching tunneling construction method. The excavation hole is about 14.50 m wide and 9.60 m high. The left hole is 4187 m long, the right hole is 4150 m long, and the tunnel has a maximum depth of 392 m. It is a deep and long tunnel. The formation of the tunnel site is mainly composed of marine facies and neritic facies carbonate rocks, clastic rocks, and continental facies clastic sedimentary rocks. With the exception of Cretaceous and Tertiary, the strata of the outcrop developed in different degrees from Triassic to Quaternary, with the highest thickness and widest distribution in the Jurassic. The lithology mainly consists of the carbonate rock including limestone, dolomite, karst breccia, pelite, and sandstone. Among them, the carbonate rock is mainly exposed to the anticline axis and exhibits a long strip, and the sandstone and pelite rocks are mainly exposed to the two limbs of anticline.

When the right-line tunnel was excavated to construction mileage YK9 + 268, it was proved through advanced drilling that there was a water-bearing cave in front of the tunnel face, and there would be a risk of water inrush if the excavation continued. Through further detection, it was inferred that the height of the cave was about 5 m, the depth was about 10 m, the width was about 15 m, and the filling material in the cave was mainly water. After several days of heavy rainfall, the water inrush from the tunnel face was aggravated. Strands of water appeared at the top of the tunnel, the left side wall, the arch, and the floor. The scattered water appeared on the right side arch. The left side and vault were more permeable; the instantaneous water inflow was about 7000 $m^3$/d, and the total water output reached 4000 $m^3$/d.

## 3. Methods

### 3.1. Model Test of Water Inrush

#### 3.1.1. Test System

The test system must provide the geological environment required by the model test, and control the test process. The test system mainly includes the ground stress loading system, the water pressure loading system, the test box, and the multi-information monitoring system, as shown in Figure 2. The corresponding model test system was developed aiming at a geological environment with a large

buried depth and high water pressure in the test section. The overall design requirements of the test system are as follows:

(1) The ground stress loading system includes a hydraulic pump station and a jack. The peak pressure of the ground stress loading system can reach 100 t, and the system can realize a grading pressurization and maintain the pressure long-term.

(2) The hydraulic pressure loading system includes an air compressor and a holding water pump. The variation in water pressure loading is between 0–6 MPa, and the system records the hydrostatic pressure value automatically during testing.

(3) The test box is a whole frame structure with good sealing, transparent visibility, and high strength.

(4) The multi-information monitoring system is used for the real-time monitoring of the stress, strain, and tunnel displacement, and a camera is used to monitor the rupture of the surrounding rock and the process of the water inrush.

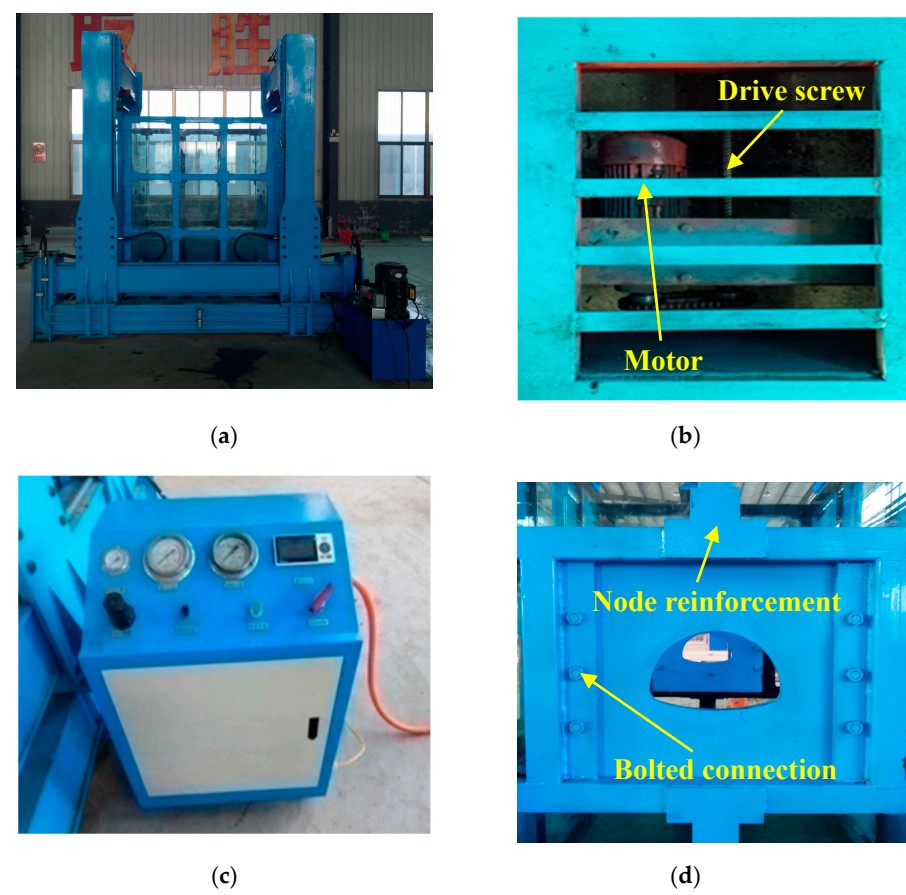

(**a**)   (**b**)

(**c**)   (**d**)

**Figure 2.** Model test system of water inrush in large buried depth and high water pressure tunnels. (**a**) Model test stand, (**b**) Driving mode of anti-force frame, (**c**) Hydraulic loading system, (**d**) Test box.

3.1.2. Similar Materials

The development of similar materials is the key to the success of the model testing of the geomechanics. A similar material to the surrounding rock that was used in the test was self-developed through Shandong University. The raw materials were sand, calcium carbonate, iron powder, cement, chlorinated paraffin, and silicone oil. According to the equations of the prototype and model, such as

the balance, geometry, physics, and boundary conditions, the similarity of each physical quantity was deduced as follows [20]:

$$C_\sigma = C_\gamma \cdot C_L$$
$$C_\varepsilon = C_f = C_\phi = C_\mu = 1$$
$$C_\sigma = C_E = C_c = C_{\sigma_c} = C_{\sigma_t}$$

In the formula, $\sigma$ (Pa) denotes the stress, $\gamma$ (N/m$^3$) is the bulk density, $L$ (m) indicates the length, $\varepsilon$ is strain, $f$ denotes the friction coefficient, $\varphi$ (°) is the friction angle, $\mu$ is the Poisson ratio, $E$ (Pa) is the elastic modulus, $c$ (Pa) indicates the cohesion, $\sigma_c$ (Pa) is the compressive strength, $\sigma_t$ (Pa) is the tensile strength, and $C$ denotes a similar measurement corresponding to them. It can be seen that the similarity ratio of all the physical quantities can be obtained through a combination of the geometric similarity ratio and the severe similarity ratio. According to the geological prospecting data and similarity principle, the physical and mechanical parameters of the surrounding rock and similar materials of the test section were obtained, as shown in Table 1.

**Table 1.** Physical and mechanical parameters of surrounding rock and similar materials.

| Medium | Gravity/ (kN/m$^3$) | Compressive Strength/ MPa | Tensile Strength/ kPa | Elastic Modulus/ GPa | Cohesion Force (C)/ MPa | Internal Friction Angle ($\varphi$)/° | Poisson Ratio ($\mu$) |
|---|---|---|---|---|---|---|---|
| surrounding rock | 26.6 | 34.36 | 888 | 46.65 | 1.47 | 41.8 | 0.12 |
| similar materials | 24.6 | 0.64 | 16.39 | 0.86 | 0.027 | 41.8 | 0.12 |

Note: The physical and mechanical parameters of the surrounding rock are checked through a geological engineering explanation of the rest-horse tunnel. Mileage pile number: Yk9 + 266~Yk9 + 296. The physical and mechanical parameters of similar materials are the average of 10 groups of specimens.

After a large number of proportioning tests, similar materials that meet the requirements of this test were made, and their weight ratios are shown in Table 2. In addition, gypsum has the characteristics of fast hardening and a strong bonding capability, and the compressive strength is within the range of 3–5 MPa. Referring to previous experience, gypsum was selected as a similar material as the initial shotcrete [21].

**Table 2.** Proportions of similar material for surrounding rock.

| Raw Material | Sand | Calcium Carbonate | Iron Powder | White Cement | Chlorcosane | Silicone Oil |
|---|---|---|---|---|---|---|
| ratio | 1 | 0.08 | 0.1 | 0.13 | 0.12 | 0.032 |

### 3.1.3. Design of Test Conditions

A geometric similarity ratio of 1:50 was chosen, and the tunnel section is a curved wall with three circular parts. The excavation width and height of the main tunnel are 14.50 m and 60 m, respectively, and that of the corresponding section of the model are 29 cm and 19.2 cm. The excavation location is located in the middle of the model body, which satisfies the boundary conditions. The filling height of the similar material is 180 cm, which is about 80 cm higher than the tunnel roof, and can simulate an actual buried depth of 40 m, and the average buried depth of the test section is 235 m. The remaining load is provided by the ground stress loading system, and the vertical stress is about 104 kPa. The method of pre-overload pressurization was adopted in the test. The stress state of the surrounding rock was formed over hundreds of millions of years, and it is difficult to simulate the model material through short-term loading. Therefore, the model confining pressure started with 2.5–3 times the design pressure loading, which accelerated the creep of the model material. After the model was stabilized, the pressure was lowered to the design confining pressure to start the test. A ball-type cave with a radius of 14 cm was preset in front of the tunnel, and the radius was about equal to the width of

the excavation. The distance between the cave center and hole was 90 cm. The cave water pressure was 25 kPa, and the water pressure remained unchanged during the test. The model was excavated after the simulated stress and water pressure were stabilized. The design of the model test condition is shown in Figure 3.

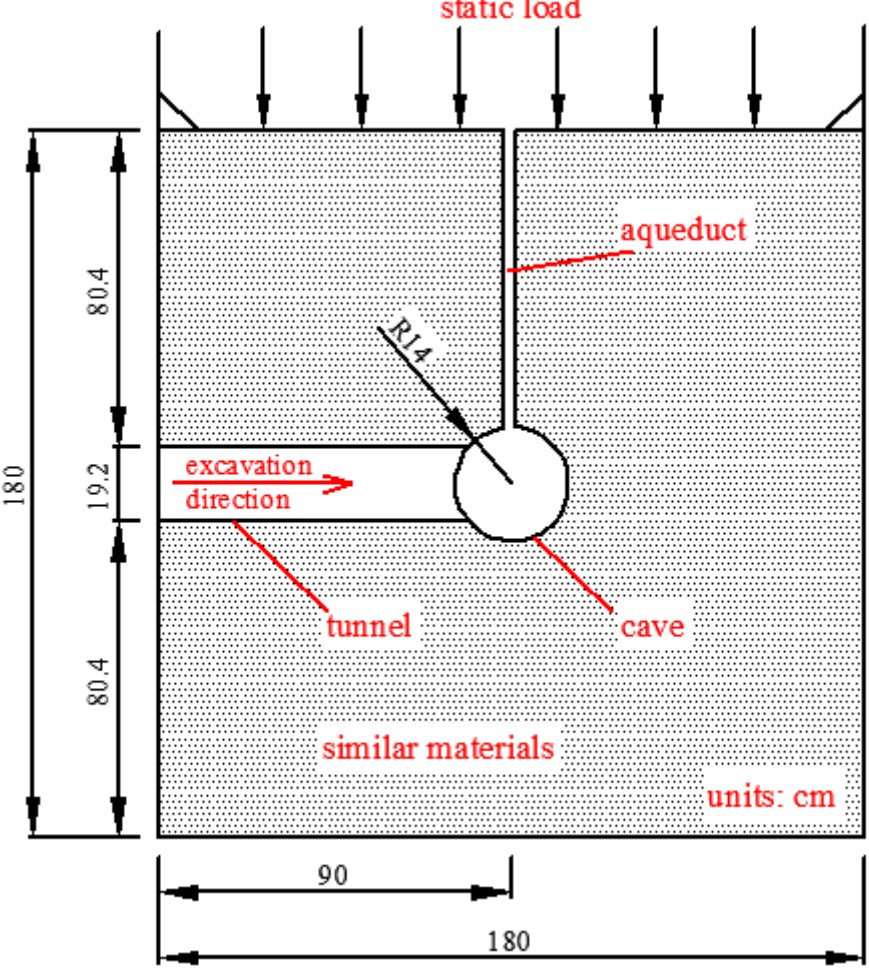

**Figure 3.** Model test condition design.

### 3.1.4. Monitoring Scheme Design

Four monitoring sections (I–IV) were arranged along the tunnel axis, and their distances between the monitoring sections and the cave center were 56 cm, 42 cm, 28 cm, and 18 cm, respectively. Points at representative locations were selected as monitoring points, such as outside the vault, the arch foot, and the arching bottom of the contour line. The monitoring information includes the stress, strain, displacement, and seepage pressure of the surrounding rock, and the corresponding monitoring elements are the miniature pressure box, strain brick, fiber grating sensor, and osmometer. Each of the four monitoring elements is referred to as a group, and in view of the symmetry of the left and right arches, the two arch feet share a group of monitoring elements. As a result, there are 16 monitoring points and 48 monitoring elements. The range and accuracy of the different monitoring elements are different, and the spacing of each component of the same monitoring point is reasonable and without interference. The positions of the monitoring points are shown in Figure 4.

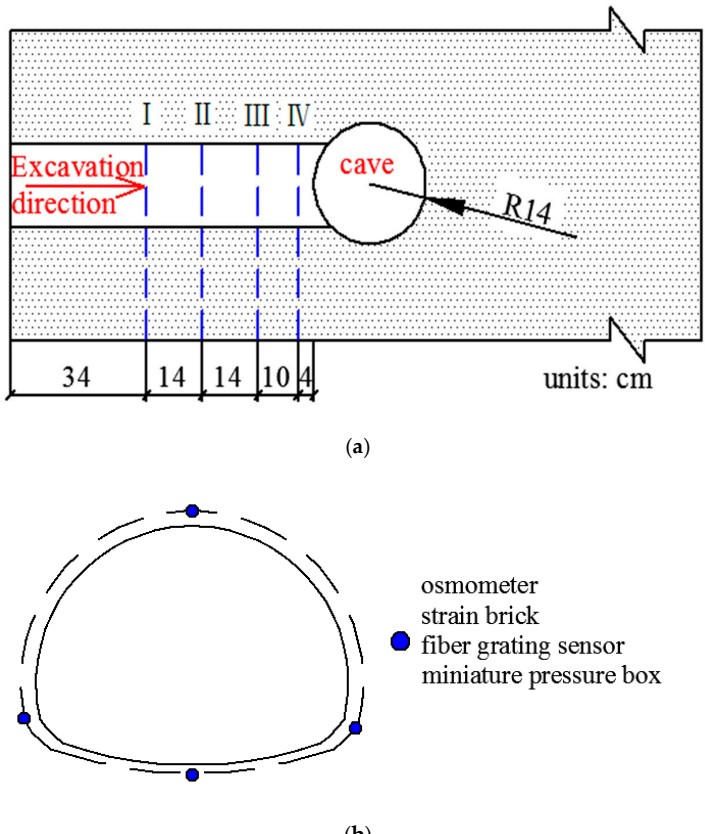

(**a**)

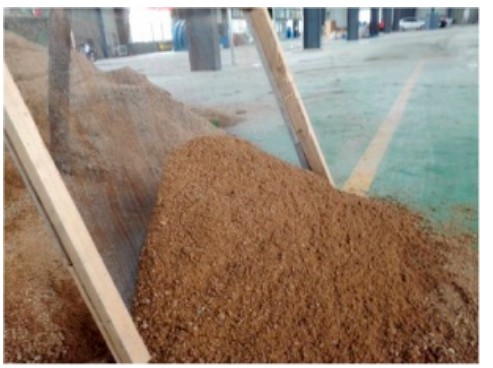

(**b**)

**Figure 4.** Sketch of monitoring section location and monitoring point location. (**a**) Sketch of monitoring section location, (**b**) Sketch of monitoring point location.

### 3.1.5. Fabrication and Excavation of the Model Body

The model body represents the location where the experimental phenomena are produced and evolved, which is directly related to the accuracy of the test results. To ensure the compactness and uniformity of the model, layered filling and artificial compaction were adopted.

The concrete steps were as follows: the screening of raw materials, mixing of raw materials, filling of similar materials, laying of monitoring elements, presetting of the karst cave, and prefabrication of the water pipes, as shown in Figure 5.

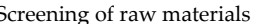

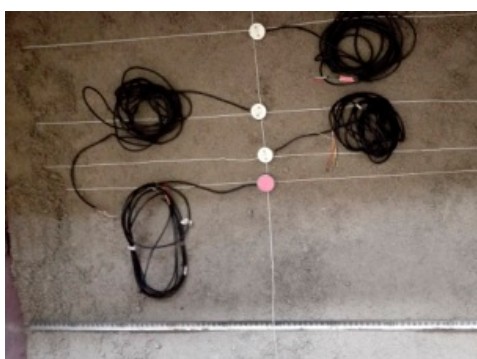

Screening of raw materials                    Component embedment

**Figure 5.** *Cont.*

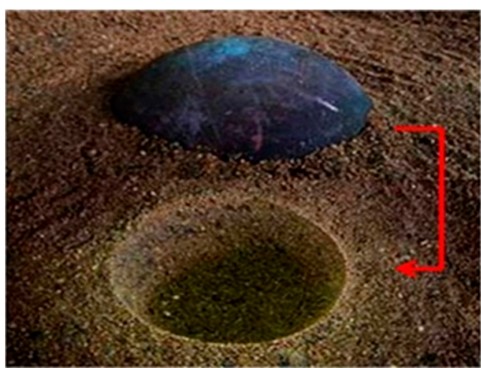
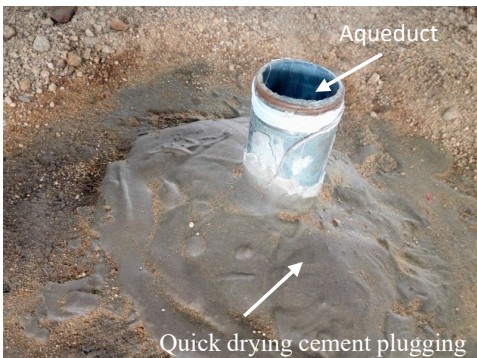

Precast karst cave          Prefabricated aqueduct

**Figure 5.** Model-making process.

After the model body was completed, ground stress and water pressure loading were carried out to simulate the large buried depth and high water pressure geological environment. The ground stress and water pressure loading steps on the model body are as follows:

(1) Ground stress overload pressure. The test used three times the initial ground stress to carry out overload pressurization, i.e., 3 $\gamma$h = 312 kPa. Step-increased loading was applied, and the loading gradient was 20 kPa. The action time of each stage was 10 min until the pressure was increased to the calibration value;

(2) Maintain ground stress overload loading. Continue overloading for 15 to 20 days. The model body was continuously creeped to reduce the porosity of similar materials. Until the model body was air-dried and the deformation was stable;

(3) Ground stress was loaded according to the design value. The ground stress value was reduced to the ground stress at the actual buried depth, i.e., $\gamma$h = 104 kPa;

(4) Water pressure loading. Step-increased loading was applied and the loading gradient was 5 kPa. The action time of each stage was 10 min. The water pressure was kept stable after being loaded to the calibration water pressure. The water pressure loading was stopped after the water inrush occurred.

A manual drilling method was used for excavation, and the bench method was used for construction. The total length of the excavation was about 76 cm. The height of the upper bench was 11 cm, and that of the lower bench was 8.2 cm. Each cycle footage in the ordinary surrounding rock section was 4 cm (0–48 cm), and was 2 cm in the surrounding rock section affected by karst caves (48–76 cm). As the supporting principle, water inrush in front of the tunnel face during actual construction was not applied for the lining structure, and thus there was no support structure in this test, and only a simple support was made in the tunnel using gypsum. When the monitoring data were stable, the next cycle was repeated.

*3.2. Numerical Simulation of Water Inrush*

3.2.1. Geometrical Model and Boundary Condition

The cave model in front of the tunnel was established using the finite element software ANSYS (ANSYS Inc., Pittsburgh, PA, USA), and the model was then introduced into the finite difference software FLAC$^{3D}$ to calculate the excavation. Considering the actual working conditions, the boundary effect, and the location of the karst cave, the following model was established (Figure 6). In the model, a part of the karst cave was divided into a free mesh with an axial length of 12 m. The excavation part of the tunnel was mapped through mesh stretching, and the axes were stretched by 34 m in the front and rear directions, and the grid spacing was 1 m. The model size was 60 m × 80 m × 60 m, and the total number of units was about 13,000. The coordinate origin was located in the center of the cave, in which the horizontal direction was the x-axis, the tunnel axis was the y-axis, and the vertical direction

was the z-axis. The tunnel span was 12 m, the cave diameter was 11 m, and the excavation distance was 34 m. The excavation method was the bench method. The footage of each excavation was 2 m in the first 22 m, and 1 m in the later 12 m. There were 23 excavation steps altogether. The boundary condition of the model was realized by applying the following displacement constraints: a horizontal direction displacement constraint to the left and right surfaces of the model, an axial displacement constraint to the before and rear surfaces, and a vertical direction displacement constraint to the lower surface. The upper surface was a free boundary, and the vertical load corresponding to the actual buried depth was applied. The seepage boundary of the six surfaces was a permeable boundary, and the pore water pressure was applied according to the actual pressure. The cave wall and the tunnel face were also provided with a permeable boundary, which corresponded to the model test. No support structure was provided during excavation.

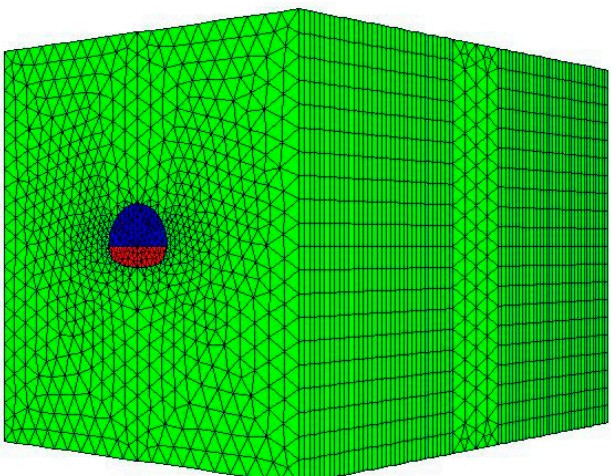

**Figure 6.** Meshing of the model.

### 3.2.2. Ground Stress and Water Pressure Loading

To prevent a yield element during the process of ground stress loading, the initial ground stress was calculated using an elastic solution method; that is, the material constitutive model was set to the elastic model, and the volume modulus and the shear modulus were obtained. After the initial ground stress calculation was completed, the model parameters were obtained using the Mohr–Coulomb model. The parameter values are shown in Table 1. The seepage mode was opened during hydraulic loading, and the water level was set on the surface of the model. The water pressure of the karst cave was 1.24 MPa, and an isotropic permeable model was adopted. No consideration was given to the compressibility of the soil particles. The osmotic coefficient of the surrounding rock was shown to be $1 \times 10^{-12}$ m$^2$/pa s, the density of the water was $1 \times 10^3$ kg/m$^3$, the bulk modulus of water was $2 \times 10^9$ pa, and the porosity of the surrounding rock was 0.35.

## 4. Results and Discussions

### 4.1. Analysis of Phenomenon

#### 4.1.1. Model Test Phenomenon

For an accurate description, the upper bench and lower bench in the same section were all called a single excavation step. The tunnel face was dry during the initial excavation, with no seepage or leakage, which indicates that the model body was well sealed. When the model was excavated to the 12th step, the water seeped at the top of the tunnel, and it is speculated that tiny fissures were produced at the inner part of the rock, and local water seepage occurred at the tunnel face. The water seepage velocity of the tunnel face then increased, from a point-like drip to a line-like drip, which is

inferred to be the process of the spreading fissure. The water seepage volume of the tunnel face surged into a stock gush when the model was excavated to the 21st step, which is speculated to be from the water inrush from the karst cave caused by the coalescence of the water flowing fractures. The process of water inrush can be summed into three stages by analyzing the phenomenon of the model test: the initiation of a water flowing fracture, the extension of the water flowing fracture, and the transfixion of the water flowing fracture. The experimental phenomena are shown in Figure 7.

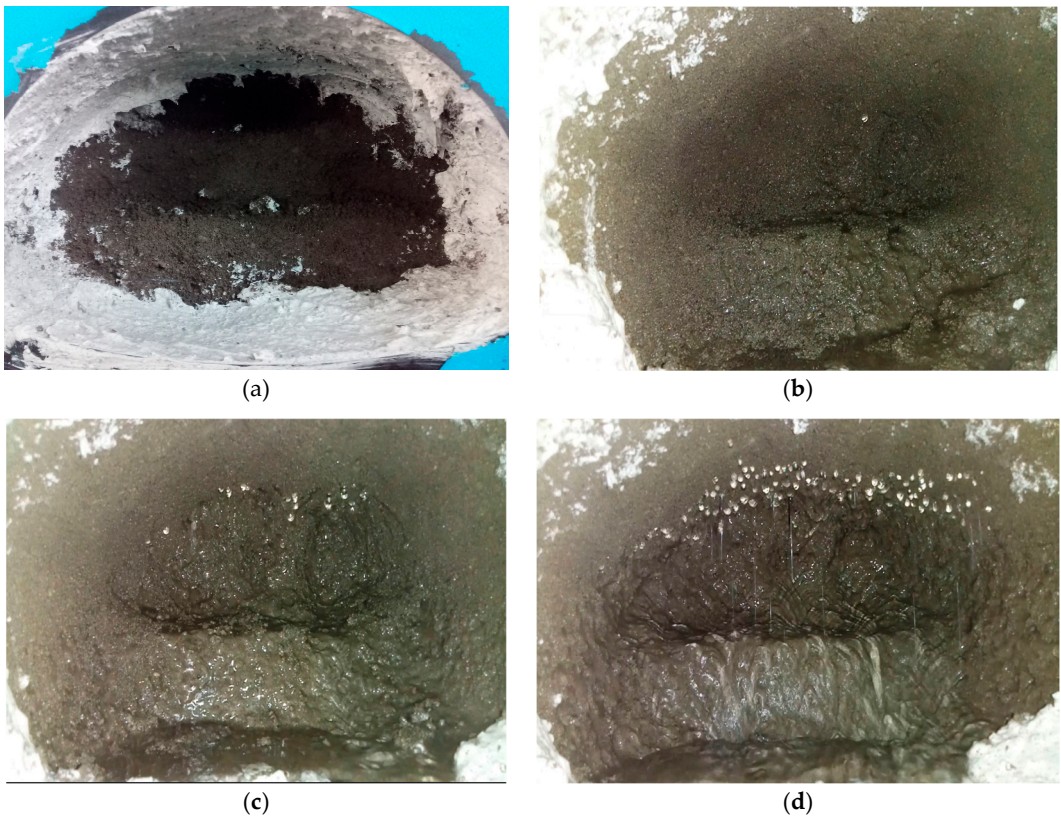

**Figure 7.** Model testing phenomenon, (**a**) Dry tunnel face, (**b**) Local seepage in the tunnel face, (**c**) Large area seepage of the tunnel face, and (**d**) Water inrush from the tunnel face.

### 4.1.2. Numerical Simulation Phenomenon

The water inrush process was analyzed from the surrounding rock plastic zone and pore water pressure. To show the characteristics of these two aspects, the location of the plastic zone was x = 30, and the location of the pore water pressure section was x = 30 and y = −6. After the calculation of water pressure balance, a high valued circle of the plastic zone and the pore water pressure were found around the cave, and the plastic zone was mainly a shear failure. The area of the plastic zone of the karst cave in the first through the 17th excavations was basically unchanged, and the area of the plastic zone in front of the tunnel face was enlarged. Corresponding to the model test, this is the initiation process of a water flowing fracture. The plastic zone of the 17th and 18th excavations was suddenly connected, and the radius of the high valued circle of the pore water pressure increased. Corresponding to the model test, this is the process of the extension of the water flowing fracture. The area of the plastic zone in the 18th through the 23rd excavations continued to expand, the main form of the plastic zone failure of the palm face was a shear failure, and some tensile failure units appeared. The radius of the high valued circle of the pore water pressure continued to increase. It is speculated that the transfixion of the water fissures during the process led to water inrush. The changing process of the plastic zone and the pore water pressure are shown in Figures 8 and 9, respectively.

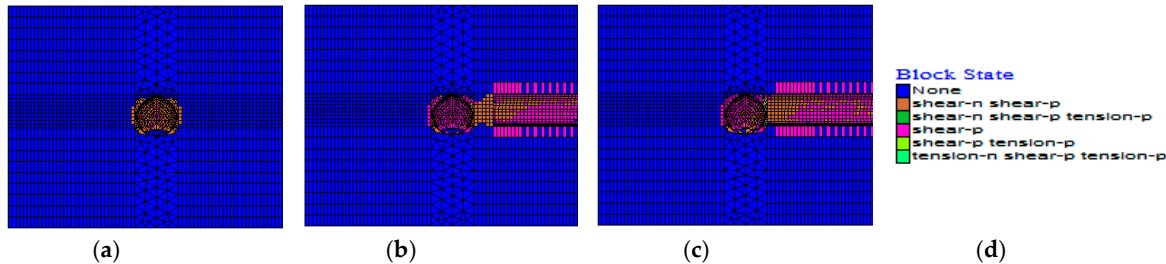

**Figure 8.** Plastic zone of the surrounding rock, (**a**) no excavation, (**b**) 18th excavation, (**c**) 23rd excavation, (**d**) Legend.

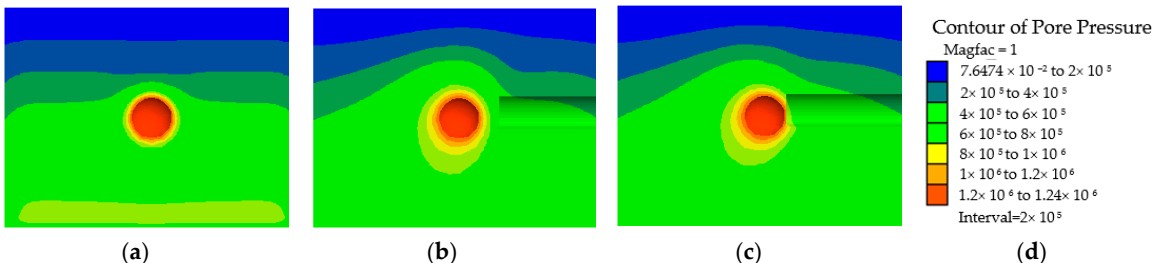

**Figure 9.** Contour of the pore water pressure. (**a**) No excavation, (**b**) 18th excavation, (**c**) 23rd excavation, (**d**) Legend.

### 4.2. Analysis of Multi-Field Information Law

4.2.1. Comparative Analysis of Multi-Field Information of Arch Ring

The model test and simulation results do not match exactly in terms of magnitude. It proves that the breakthroughs in the model test and numerical simulation are needed. For example, the manual drilling method or mechanical excavation is always applied in the model test, while the model "NULL" is usually adopted to simulate the method of blasting excavation in numerical simulation. The stress release process of the two methods is different, so it will definitely lead to inconsistent results. It is necessary to consider a viscosity similarity in the model test, but no suitable material can be found to simulate the water. The hydraulic conductivity between the model test and the project was simulated in the test, but this is obviously not enough. However, this problem does not exist in the numerical simulation, and the fluid factor can be adjusted according to the actual situation. In addition, there are other factors that will increase the diversity, such as the material preparation and detection methods. However, considering the advantages of model tests in analyzing damage phenomena and the advantages of numerical simulations in the process of variation, the combined research of model tests and numerical simulations is beneficial.

In order to facilitate the analysis of the results, the data of the model test and the numerical simulation are normalized and compared.

(1) Displacement analysis

As shown in Figure 10, the results of the model test and the numerical simulation are consistent with the change in the vault displacement, which shows a continuous settlement. With the continuous advancement of the tunnel face, the vault successively produced an early settlement, an instantaneous settlement, and a late settlement, and the instantaneous settlement was at maximum, which indicates that the excavation disturbance has a significant influence on the stability of the surrounding rock. Compared with the monitoring sections I and II, the early settlement was produced in monitoring section III, which indicates that the existence of the water-bearing karst cave makes the tunnel surrounding rock more susceptible to buckling deformation, the difference being that the results of the model test increased a second time in the tunnel water inrush, with monitoring section III being the most obvious, and the final settlement being larger than monitoring sections I and II. It is inferred

that this phenomenon was affected by the scouring of the karst cave gushing water, which indicates that the water inrush aggravated the late settlement rate of the surrounding rock.

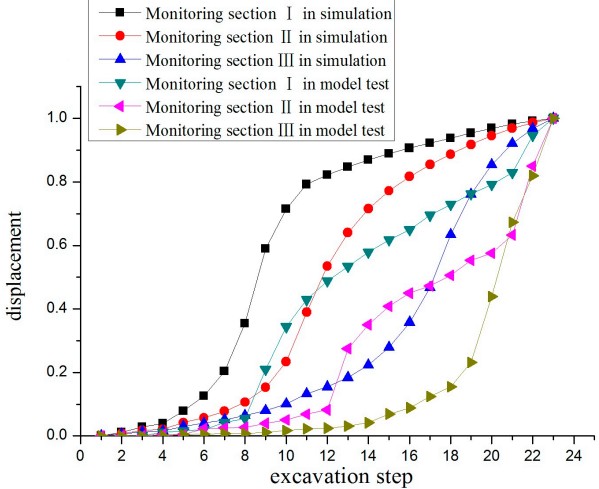

**Figure 10.** Variations in displacement during excavation process.

(2) Stress analysis

As shown in Figure 11, the results of the model test and numerical simulation are consistent with the change in vault stress, which showed a slow increase initially, and then a sharp decrease. The stress of the surrounding rock gradually increased during the initial stage of excavation, which exceeded the initial stress level, indicating that the stress concentration consistently led to damage and failure of the surrounding rock. When the excavation surface was excavated to the monitoring section, the stress value of the surrounding rock fell steeply owing to the excavation unloading action, which indicated that the damage and failure of the surrounding rock was aggravated. When the water inrush occurred, the stress of the surrounding rock was not obvious, which indicated that the water inrush no longer affected the stress of the stabilized surrounding rock. The maximum stress of monitoring section III was less than that of monitoring section II, which shows that the surrounding rock near the karst cave was weakened by the erosion of water, the strength of the rock mass was reduced, the accumulation of energy of the surrounding rock was decreased, and the stability of the rock mass was insufficient. There was also a risk of collapse after excavation.

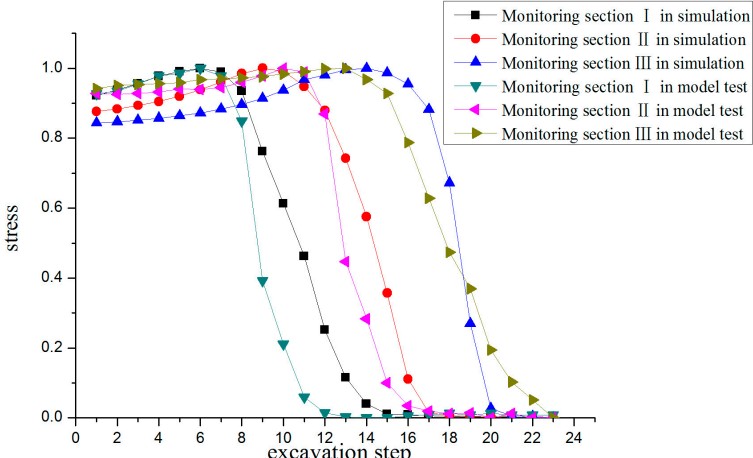

**Figure 11.** Variations in stress during excavation process.

(3) Seepage pressure analysis

As shown in Figure 12, the results of the model test and numerical simulation are similar with the trend of the seepage pressure, and the performance decreased initially, and then increased. The seepage pressure of the surrounding rock gradually decreased during the initial stage of excavation, which indicates that small fissures were produced in the rock and soil body during the stress redistribution, and that the pore water was discharged along the fissures, which slowed down the seepage pressure. The closer the monitoring section was to the karst cave, the smaller the seepage pressure drop rate, and the greater the minimum seepage pressure. This is more obvious in the model test results, which shows that the existence of the karst cave changed the original seepage field of the tunnel surrounding rock. After the excavation face passed through the monitoring section, the pore water pressure gradually increased, which was mainly caused by excavation interference or supplemental fissure water factors. When the water inrush was initiated, the seepage pressure increased a second time, which is obvious in the model test, indicating that the water flowing fissure expanded continuously, and a water-guiding channel was formed.

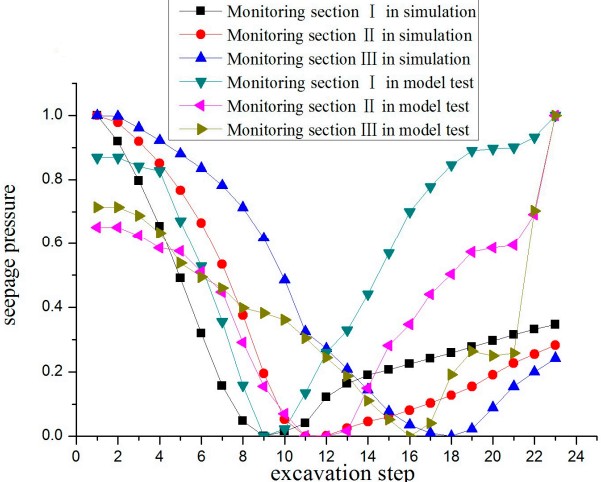

**Figure 12.** Variations of seepage pressure during the excavation process.

Through the above analysis, it was shown that the results are basically consistent between the model test and the numerical simulation, and the vault displacement is characterized as a continuous subsidence. The surrounding rock stress increased at first and then decreased, which is manifested as stress concentration and release. In addition, the seepage pressure decreased initially and then increased.

### 4.2.2. Comparative Analysis of Multi-Field Information in front of Karst Cave

In the preceding analysis, the information of the different physical fields of the arch ring is basically consistent between the model test and the numerical simulation. The difference is that the displacement and seepage pressure of the karst cave were increased at different degrees in the model test, and the data from the numerical simulation showed no clear change. To determine whether karst water inrush was produced in the numerical simulation, the physical field information in the front of the karst cave was further analyzed, that is, the displacement, stress, and seepage pressure data of the tunnel axial unit were extracted. During the numerical simulation, the plastic zone of the surrounding rock started to transfix at the 18th step, and thus the extraction unit number is 81,161–81,165, and the node number is 33,221–33,225. In addition, to consider the variations in the test data synthetically, the linear function conversion method was used to map the physical information, such as the stress, displacement, and seepage pressure of monitoring section IV to a range of 0 to 1, and draw in the same coordinate system.

(1) Analysis of model test data

According to Figure 13, compared with the first three sections, the early settlement of monitoring section IV appeared ahead of schedule. The vault sinking volume was steep during the 19th step, the stress concentration phenomenon was not obvious, and the amount of stress decreased sharply at the same time. The seepage pressure decreased gradually during the 17th step and increased rapidly during the 21st step. This shows that the strength of the surrounding rock is low and its stability is poor, which makes the water inrush have a sudden and transient nature. Corresponding to the test phenomena, the water-guiding channel was transfixed and water inrush occurred at about the 21st step, and the distance between the cave front wall and the tunnel face was about 10 cm. It was estimated that the minimum safety thickness in front of the tunnel face is about 5 m.

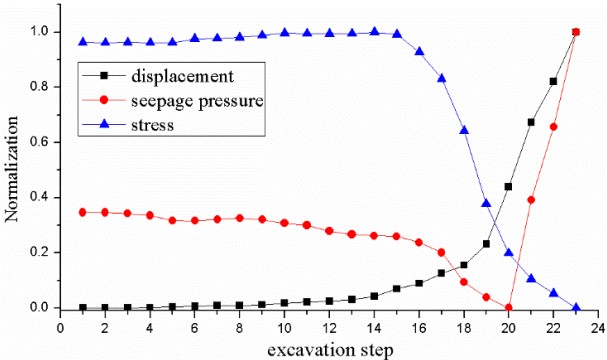

**Figure 13.** Normalized curve of monitoring section IV.

(2) Analysis of numerical simulation data.

The model test shows that the water yield suddenly increased when the water inrush occurred in the karst cave, and it is presumed that the flow velocity increased. The change in pore water pressure rate is the main basis of the fissure transfixing the water inrush during the numerical simulation. The pore water pressure rate is defined as $v = p_{i+1} - p_i$. It is shown in Figure 14 that the slope of the pore water pressure rate during the first through 18th steps is negative, and the slope can be considered an approximately fixed value. The pore water pressure showed a gradual change, which is the initiation and propagation process of water flowing fractures. The slope of the pore water pressure rate changed from positive to negative during the 19th through 23rd steps, the slope was abrupt, and the pore water pressure value increased sharply. In addition, the displacement and stress of the axial element were both mutated during the 19th step. Therefore, it was determined comprehensively that the water guiding channel was transfixed at the 19th step, and the distance between the excavation and karst cave is 4.5 m.

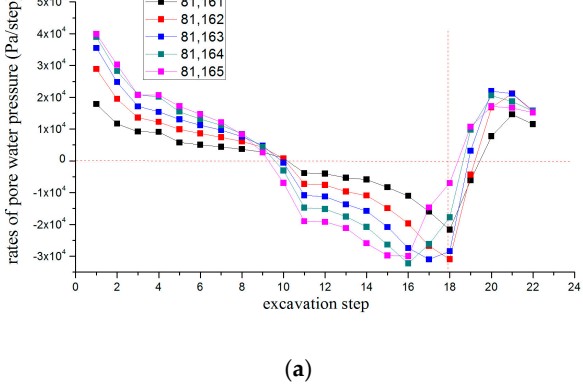

(**a**)

**Figure 14.** *Cont.*

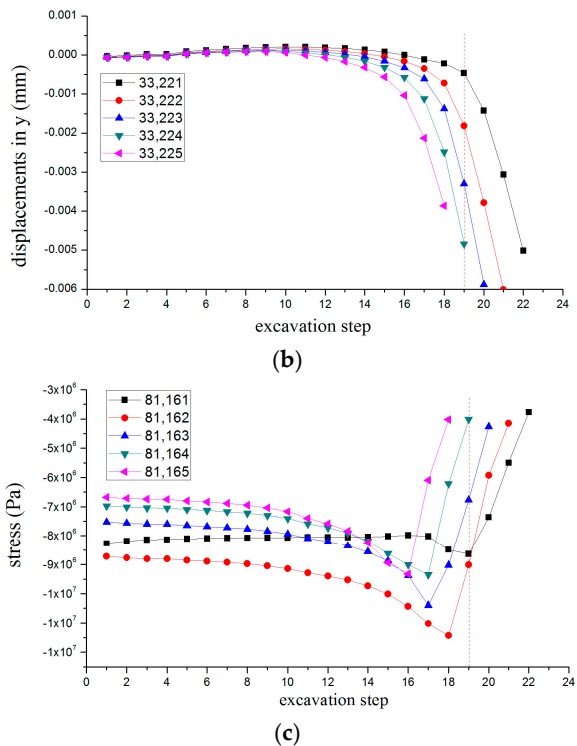

**Figure 14.** Multi-field information change curve from numerical simulation. (**a**) Rates of pore water pressure, (**b**) Displacements in y, (**c**) Stress.

The above analysis shows that the experimental phenomena and data characteristics in the model test indicate that the water inrush occurred during the 21st step, and the thickness of the water-resisting rock of the karst cave is 5 m. The concept of the pore water pressure rate was introduced in the numerical simulation. The results indicate that the water inrush occurred during the 19th step according to the characteristics of the seepage pressure, displacement, and stress, and that the thickness of the water-resisting rock of the karst cave is 4.5 m. The two simulation results are quite close.

Through comprehensive analysis of model tests and numerical simulation results, it is found that the changes of the laws of physical field are basically the same, and the results are reliable. However, there was a slight difference in the extreme of the data and the time at which it occurred. There are two possible reasons. First, there were some differences in the excavation methods in the model tests and numerical simulations. The manual excavation method was adopted in the model test, and the excavation process was disturbing to the surrounding rock. Second, there were more external factors in the model test, and the numerical simulation environment was more ideal. Boundary conditions and material properties were simplified in numerical simulations.

## 5. Conclusions

Aiming at a series of prominent problems during the process of tunnel construction in a karst area, in this study, a model water inrush test system was developed for large buried depth and high water pressure tunnels. Based on the Xiema tunnel of the "one horizontal line" project of the Chongqing Expressway System (Xie Ma interchange–Cai Jia interchange section), a model test of the flood-change evolution in front of a tunnel filled with water was carried out. A model of the catastrophic evolution of water inrush from a water-filled karst cave in front of a tunnel was also developed, and a numerical simulation of the selected test section was conducted. The results of a comprehensive model test and the numerical simulation are summarized as follows:

(1) The model test summarized three stages of water inrush in a karst cave as the initiation of a water flowing fracture, an extension of the water flowing fracture, and a transfixion of the water

flowing fracture. The numerical simulation explains the various stages from the surrounding rock plastic zone and pore water pressure: the main cause of the water inrush in a high-pressure karst cave is an expansion of the excavation fissure and an infiltration of a high-pressure water in the karst cave; in addition, a numerical simulation showed that amount of plastic area increased, and the radius of the high valued circle of the pore water pressure increased.

(2) The changing laws of the multi-field information of the arch ring used in the model test and the numerical simulation were basically consistent. The vault displacement increased continuously, the instantaneous settlement was the largest, and an early settlement was produced near the karst cave section. In addition, the stress increased slowly and then decreased sharply, which shows the stress concentration and release in front of the excavation face. The seepage pressure first decreased and then increased, which corresponds to the initiation, propagation, and transfixion process of water flowing fractures.

(3) A numerical simulation was used to further analyze the unit ahead of the karst cave. It is considered that the water inrush occurred during the 19th step, and the excavation face was 4.5 m away from the cave. In the model test, the water inrush occurred during the 21st step, the excavation face was 10 cm away from the cave, and in the actual project, this distance is equivalent to 5 m. These two simulation results are close to each other and have a certain amount of reliability, which will provide theoretical guidance for the prevention of the thickness of the surrounding rock in front of the tunnel face in similar engineering projects.

**Author Contributions:** W.Y. and S.S. conceived and designed the experiments; H.W. and R.D. performed the experiments; L.B. and M.W. analyzed the data; L.L. contributed reagents/materials/analysis tools; W.Y. and Z.F. wrote the paper.

**Funding:** This research was funded by [National Natural Science Foundation of China] grant number [51879148, 51479107], [Shandong Postdoctoral Innovation Project Special Foundation] grant number [201502025], [Natural Science Foundation of Shandong Province, China] grant number [ZR2018BEE045], and [China Postdoctoral Science Foundation] grant number [2018M630780]. And The APC was funded by [51879148].

**Conflicts of Interest:** The authors declare no conflict of interest.

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
