# Peer review of "Analysis on Water Inrush Process of Tunnel with Large Buried Depth and High Water Pressure"

_processes, doi:10.3390/pr7030134_

Round 1

Reviewer 1 Report

The manuscript proposes an analysis of the phenomenon of water inrush in underground tunnel excavations, by making comparison between a laboratory test model conceived to recreate the original setting through use of materials with similar behaviour, and numerical simulations based on a finite-element method. The results aim at offering some guidance for the protection of underground excavations from hazards of catastrophic water inrush by furnishing indicative rock layer thickness needed for the protection of the cave.

Unfortunately, I believe this work is not suitable for publication in the present form. While the study centainly has its relevance, especially in view of its results, its biggest issue is the poor description of the experimental and simulation settings. I believe they both need to be described adequately in order to give readers the ability to understand how the analyses have been undertaken, and thus give the right significance to the proposed discussion and results. 

The introduction (Section 1) is overall well written and informative, as it gives a basic context of the field of knowledge, although I feel that in the ending paragraphs the authors should describe and highlight a bit more the main innovations of their work. I suggest authors follow the standard format for citations by putting the year right after the name of the authors of the cited work: i.e., at line 60, put the (2016) right after Li et al..

Sections 2 and 3 are, in my opinion, a major point of concern, since they do not describe the overall setup of the study adequately. 

The Engineering Background should aim at describing clearly the main features of the tunnel-cave system, but I found it rather confusing. I believe some brief lines of context about the main uses of the tunnel and the construction techniques would be useful.

As the authors describe the background, several switches between present (L79, "... there is a water-bearing cave...") and past (L89, "...it was inferred that...") verbal tenses are made, leaving the reader wondering if the tunnel has been already completed or if it's a work in progress. I suggest the authors stick to a verbal tense throughout the whole description.

Some of the terminology used is not explained. As an example, the authors describe the grade of the surrounding rock as III grade (L84), but never explain what this actually means, the scale to which this assumption is referred (III grade with respect to what?). I think a reference to the classification scale used in this assumption would be a welcomed addiction. Some other nomenclature is a bit obscure, e.g. "stake number" (L79) and "femoral gushing water" (L85).

A source of confusion in L94-95: the instantaneous water inflow should be in cubic meters over a specific unit of time (like m3/s, m3/h; not only m3). However, make sure that the instantaneous water inflow and the total water output are consistent with each other.

Figure 1 offers little help in understanding the geometry of the problem and it is also incorrect (the cave area should be in square meters...).

If the authors want to make Figure 2 significant for the description of the model test they should make it clearly visible and use high quality pictures. Moreover, red labels are not so visible against the predominant blue colours of the pictures.

In Table 2, it is not clear what the ratios' meaning is (volume ratio? weight ratio? strength ratio?). 

The description of the monitoring scheme (Section 3.1.4) is also rather poor. Four monitoring section are mentioned in L170 and also recalled frequencly in the Results and Discussion, but should also be contextualized in a figure. In this regard, Figure 4 should be replaced by a 3-dimensional sketch of the test tunnel, or at least a figure with multiple views of the tunnel, in order to show the displacement of monitoring sensors in an optimal way. Also, it is not really clear why there should be 15 monitoring points, since the are four sections with 4 monitoring groups each. Please be careful in the description.

Also in Section 3.1.4, the nomenclature is puzzling. In L192 "upper step" and lower step" are cited, but their clear explanation is missing. Is "step" related to a construction phase of the tunnel? In addition, what do the author mean with "cycle footage" (L192)?

All in all, I believe the conceptual structure of the paper should be strongly improved. Please reconsider resubmitting after a thorough revision of the manuscript.

Author Response

 Great appreciation goes to the editorial board and reviewers who kindly give excellent suggestions on writing and technical issues. We have revised this paper according to the reviewers’ suggestions and tried our best to perfect it. The changes to our manuscript are highlighted by using red colored words. The answers to the comments of the reviewers are summarized in the attachment.

Reviewer 2 Report

BRIEF SUMMARY- BROAD COMMENTS

This manuscript presents the analysis on water inrush process of tunnel with large buried depth and high water pressure. In my review I present the errors that the authors should correct in order this study to be ready for publication in “Processes” journal.

SPECIFIC COMMENTS

Pages 1, Lines 2-3: In the title all the main words must begin with capital letter or with small letter. Thus, write “Analysis on Water Inrush Process of Tunnel with Large Buried Depth and High Water Pressure” or (better in my opinion) “Analysis on water inrush process of tunnel with large buried depth and high water pressure”.

Page 1, Line: Why are there * next to two names? Are there more than one corresponding authors?

Page 1, Lines 4-5: Are all the authors from the same university?

Page 1, Line 8: Don’t begin the abstract with “To”. Replace it with “In order to”, i.e., “In order to explore the catastrophic….”.

Page 1, Line 9: Finish the first sentence after the word “tunnels” and continue with “ For this reason,”, i.e., “….high water pressure tunnels. For this reason, a model test system…”.

Page 1, Lines 14-15: Write “:” after “stages” and continue with capital letter, i.e., “…divided into three stages: The production of a water...".

Page 1, Line 28: Replace “Because” with “As”, i.e., “As the integrity of the rock mass…”.

Page 1, Lines 30-31: Please check the instructions of the journal: Should you write the references like this, or maybe in numbers inside [] (i.e., [1-3], [2,3], exc.) according to their appearance in the text?

Page 1, Line 33: Delete “;” before Franjo, i.e., “(Franjo and Mario 2001;…)”.

Pages 1-2, Lines 41, 43, 47, 48, 49, 52, 56, 60 and 63: Write the year of each publication next to the names of the authors. It means, delete “(2014)” in Line 43 and write “Zhu et al. (2014)” in Line 40. Similarly continue in the other above Lines.

Page 2, Line 49: Replace ‘Based” with “based”, i.e., “…side, based on the damage mechanism…”.

Page 2, Line 67: As this text is supposed to be published in a jounral, it is not anymore just a paper. Replace “paper” with “work”, i.e., “This work describes…”.

Pages 3-4, Figure 2. In the final text, the whole Figures 2(a)-2(d) with their explanation (Lines 118-119) must be in the same page. Transfer Figures 2(a) and 2(b) in next page 4, if it is necessary.

Page 4, Lines 130-133: Give the unit of each parameter in parenthesis, f.e., “L [m]  indicates the length”, exc.

Page 4, Table 1: Each Table or Figure must be appeared immediately after their first reference in the text. Also, transfer Table 1 after Line 137. Take care that in the final text, the whole Table with its notes (Lines 144-147) must be in the same page.

Pages 5-6, Figure 3. Similarly, in the final text the whole Figure 3 with its explanation (Line 168) must be in the same page.

Pages 6-7, Figure 5. Again, in the final text all the photos of Figure 5 with their explanation (Line 184) must be in the same page.

Pages 10-11, Figure 10. The same comment: In the final text all the diagrams of Figure 10 with their explanation (Lines 288-289) must be in the same page.

Pages 11-12, Figure 12. Similarly, in the final text all the diagrams of Figure 12 with their explanation (Lines 326-327) must be in the same page.

Page 13, Line 340: You write here “Li et al 2011”, but in the references (Lines 418-420) there are two publication from “Li et al.” in 2010 and in 2016. Correct the wrong date in Line 340.

References

The references are an important part in an article. The authors have done many mistakes here,  there is something wrong in almost every publication. They have to check again for each publication and write correct the names of the authors, the title of each work, the name of the journal and the numbers of volume and page. I present some of these mistakes:

Pages 15-16, Lines 417, 425-426, 435, 437 and 448-449: Are the DOI necessary? Usually it is enough just to give the other details of each publication. Chech the instructions of the journal and if they are necessary, then add the DOI in all the publications.

Page 15, Line 410: The names of the authors are not written correctly. Replace here “Franjo S, Mario W” with “Sumanovac F, Weisser M”. Correct it also in Line 33 of page 1 (it is “Sumanovac and Weisser”, not “Franjo and Mario”).

Page 22, Line 411: You give wrong numbers of volume and pages. Replace “7:13-18” with “47(1):13-28”.

Page 15, Line 412: Again the names here are not totally correct. Replace “Gavrovsck F, DreybrodtW” with “Gavrovsek F, Dreybrodt W”

Page 15, Line 414: Another wrong name. Replace “Jeannian” with “Jeannin”.

Page 15, Line 415: You write wrong number of pages. Replace “241:211-221” with “241:177-193”.

Page 15-16, Lines 416, 418, 420, 424, 436, 441 and 445. It is the first time that I see in the references of an article someone to write “et al”. In each publication you must write the names of ALL the authors. Please do this in the above Lines.

Page 15, Lines 416-417. Don’t begin every word of the title with capital letter…Write “The mechanism and numerical simulation analysis of water bursting in filling karts conduits”.

Page 16, Line 421: Don’t write the whole name of the journal, follow the abbreviation of it. Also replace “Applied Ocean Research” with “Appl Ocean Res”.

Page 16, Lines 424-426: You forgot to write the name of the journal and the numbers of volume and pages…Also, write before the DOI “Environ Earth Sci 75(1):11”.

Page 16, Line 434: The date of this publication is wrong. Replace “(2017)” with “(2018)”.

Page 16, Lines 434-435: Where are the numbers of volume and pages? Add before the DOI “36(1):95-104”.

Page 16, Lines 436-437: Similarly, add before the DOI “75(2):1450”.

Page 16, Lines 441-442 and 445-448: Similarly with my previous comment No. 28, don’t begin every word of the titles with capital letter

Page 16, Line 443: The name of the second author is wrong. Replace “Anthony TC” with “Goh ATC”.

Page 16, Line 446: Add the numbers of volume and pages, i.e., write before the DOI “36(4):508-519”.

Page 16, Line 448: Similarly, add the numbers of volume and pages, i.e., write before the DOI “37(1):185-195”.

Page 16, Line 450. This article has a third author that you don’t refer. Also, write “Zhu B, Wu Q, Yang JW, Cui T (2014)…”

I believe that if the authors follow ALL my suggestions, this paper could be suitable for publication in “Processes” Journal.

 I would like to check it one more time before the final publication.

Author Response

(The authors gave the same response as above.)

Reviewer 3 Report

The paper aim to explore the catastrophic evolution process for karst cave water inrush in large buried depth and high water pressure tunnels. For this purpose, the Authors present a model based on Xiema tunnel where the experimental section is simulated via a finite element method. The comparison between the obtained numerical outcomes with a proposed model test system show a good matching. Title and content of the paper are according to the scope of the journal. The whole structure is well-written with the used methods adequately described and the results clearly presented that could provide a guideline to evaluate the evolution process for the karst cave water inrush. I suggest to the Authors to better highlight and immediately clarify in the Introduction section the purposes of the study also for a profitable use of the obtained results.

Author Response

Sincerely thank you for your evaluation and recognition of our work.

Round 2

Reviewer 1 Report

I thank the authors for addressing my suggestions. I believe the manuscript has improved since the previous version; however, I feel the Results and Discussion section still demands your attention since it received little to no correction with respect to the first part of the manuscript. 

In particular, I would ask you to make the Results figures (Fig. 10, 11, 12...) consistent with each other and more easily comparable; one major point of concern is the fact that the numbers portrayed in the (a) panels (the trial values, with reference to the test mode) are radically different than those shown in the (b) panels (the calculated values obtained through numerical simulation). As an example, see Figure 10: in panel (a) the trial values are in the range 0.0-1.4, while in panel (b) the calculated values are in the range 0-25. I understand they are referred to radically different systems (a real model test vs a numerical simulation), but I think an effort in uniforming results would be a great help in the view of making comparisons. 

Another example: in Fig. 11 the values of panels (a) and (b) are of opposite signs... why is that? Are the reference systems for the model test and the numerical simulations different? 

Please consider improving the quality of the results in the paper as it has been done with the introductory part.

Author Response

Model test is a study in which engineering problems are scaled down to a certain extent. According to the similarity theory, the values of displacement, stress and seepage pressure are amplified according to similar scales (this value is 50 in this work) to reflect the real physics information. However, even if the model test results are enlarged according to the rules, there will be a difference compared with the results of the numerical simulation. There are many reasons for this problem. For example, the manual drilling method or mechanical excavation is always applied in model test, while the model “NULL” is usually adopted to simulate the method of blasting excavation in numerical simulation. The stress release process of the two methods is different, so it will definitely lead to inconsistent results. It is necessary to consider viscosity similarity in the model test, but no suitable material can be found to simulate the water. The hydraulic conductivity between the model test and the project was simulated in the test, but this is obviously not enough. However, this problem does not exist in the numerical simulation, and the fluid factor can be adjusted according to the actual situation. In addition, there are other factors will increase diversity, such as material preparation and detection methods.

However, considering the advantages of model tests in analyzing damage phenomena and the advantages of numerical simulations in the process of variation, the combined research of model tests and numerical simulations is beneficial. In order to facilitate the analysis of the results, the data of the model test and the numerical simulation are normalized and compared. Moreover, we have redrawn Figure 10-12. See the details in revised manuscript.

In addition, the opposite sign in the original Figure 11 is only a problem set in the simulation software. The unit of the ordinate axis in the original Fig. 11(a) should be “kPa”.

Sincerely thank you for your valuable comments on our manuscript.

Round 3

Reviewer 1 Report

I thank again the authors for considering my comments. Just a comment on the abstract: the first sentence (L8-9) is inconsistent. I believe a more coherent sentence would be: "In order to explore the catastrophic evolution... and high water pressure tunnels, a model test system was developed..."